# Comparisons of Baseline Obesity Prevalence and Its Association with Perceived Health and Physical Performance in Military Officers

**DOI:** 10.3390/biology11121789

**Published:** 2022-12-09

**Authors:** Brittany S. Hollerbach, Christopher K. Haddock, Filip Kukić, Walker S. C. Poston, Nattinee Jitnarin, Sara A. Jahnke, Justin A. DeBlauw, Katie M. Heinrich

**Affiliations:** 1NDRI–USA, Inc., 1920 W 143rd Street, Suite 120, Leawood, KS 66224, USA; 2Police Sports Education Centre, Abu Dhabi Police, Karama Street, Al Manhal W14-02, Abu Dhabi 253, United Arab Emirates; 3Department of Health and Human Physiological Sciences, Skidmore College, 815 N Broadway, Saratoga Springs, NY 12866, USA; 4Department of Kinesiology, Kansas State University, Natatorium 8, 920 Denison Avenue, Manhattan, KS 66506, USA

**Keywords:** soldiers, first responders, body composition, body fat, waist-to-height ratio, body mass index

## Abstract

**Simple Summary:**

United State (US) military members have increasing obesity rates that threaten human performance optimization. This study focused on a mostly officer population attending graduate school at the US Command and General Staff College and measured obesity, fitness, and self-rated health. Almost 80% of the 136 participants were officers; 18.5–39.7% were obese depending on which measure was used, and obesity rates differed by sex. When using body fat percentage as the standard (since it directly estimates body fat), obesity was underestimated by other measurements including body mass index (a ratio of height to weight), waist circumference, and waist to height ratio. Soldiers who were obese were much more likely to fail the Army fitness test and reported lower self-rated health than non-obese soldiers. Sample obesity rates were higher than previous studies, especially for women soldiers. To optimize human performance among USA military officers, interventions are needed to lower obesity rates and improve self-rated health.

**Abstract:**

Increasing obesity rates among USA military members vary by age, pay grade, and measurement methods and threaten force fitness and readiness. Limited research has directly measured obesity among officers; those enrolled in graduate school at the Command and General Staff College (CGSC) face additional demands and increased obesity risk. This study compared obesity measurements and performance on the Army Physical Fitness test and self-rated health for a sample of mostly CGSC officers. Participants (*n* = 136, 75.7% male, 79.6% officers) completed body composition measures; BMI, body fat percentage (BF%), waist circumference (WC), and waist-to-height ratio (WHtR) estimated obesity rates and were compared for discordance. Army Physical Fitness Test pass/fail rates and self-reported perceived health were compared with each obesity classification. Obesity prevalence was 18.5–39.7%, differing by sex and measure. BMI, WC, and WHtR underestimated obesity compared to BF% at least 21% of the time. WHtR had the lowest discordance with BF%. Soldiers categorized as obese were significantly more likely to fail the fitness test and report lower self-rated health than non-obese soldiers. Overall, obesity rates among our primarily officer sample were higher than previous studies, particularly for women. Interventions to address obesity are necessary among military officers to optimize human performance.

## 1. Introduction

Like the United States (US) civilian population [1], obesity has been increasing among active-duty military personnel and National Guard/Reserve members [2,3]. Over 14% of the US Army is now considered obese according to the latest Department of Defense (DoD) Health Related Behaviors Survey (HRBS), which calculated body mass index (BMI) from respondents’ self-reported height and weight [2]. National Guard/Reserve men had higher levels of obesity compared to their active duty counterparts, veterans, and civilians [3]. However, rates of obesity are not uniform across the Army, with both increases in pay grade and age associated with higher rates of overweight and obesity [2]. Limited research has examined directly measured body composition among US Army officers, who face different challenges than enlisted soldiers [4,5].

Obesity poses a threat to national security partly through its association with immunologic dysfunction [6] as well as its negative association with optimal human performance that threaten operational readiness [7]. Each US service branch has unique fitness and weight standards [7], and it is standard practice in the Army and other branches of the military to use BMI as a measure of body composition. Soldiers failing to meet BMI standards are subjected to a tape test that includes neck and waist circumference (WC), as well as hip circumference for women [8]. WC alone is often used to indicate increased obesity-related health risks from central adiposity [9]. Besides its utility for indicating obesity, waist to height ratio (WHtR) has been proposed to better predict cardiometabolic risk (e.g., diabetes, hypertension, and cardiovascular disease) over both BMI and WC [10,11]. Yet, limited research has examined WHtR among US military personnel [12,13]. Finally, as a direct estimate of adiposity, body fat percentage (BF%) is easily measured in the field using bioelectrical impedance analysis (BIA) [14].

Among US Air Force personnel, WC and BMI underestimated obesity in relation to BF% via BIA [15]. In particular, false negative rates for WC and BMI ranged from 35% (WC in men) to 78% (BMI in women) [15]. Among active duty US military personnel, BMI in combination with WC was satisfactory at best in assessing obesity in comparison with BIA [16]. This could be attributed to the fact that BMI and WC evaluate longitudinal and transversal body dimensions as well as body mass relative to body size (kg per each m^2^ of body size) [17]. It is imperative that researchers consider assessing both anthropometric and body composition metrics to model adiposity in military fitness field studies, as relationships between these measures may vary depending on the study’s population. Further, it is important to examine under-studied populations such as officers in the military to appropriately inform tailored health promotion efforts.

Military officers enrolled in graduate school at the US Command and General Staff College (CGSC) complete a minimum of a 10-month degree program, with a variety of degree options [18]. Due to the accelerated nature of the program, officers are under considerable stress to perform well academically while also balancing personal and family demands [19]. At the time of this study, all officers were required to pass the Army Physical Fitness Test (APFT) prior to and upon completion of the academic program in order to participate and subsequently to graduate [20]. Yet, physical training requirements have varied over time for CGSC students from no official requirements to 5 days per week of group physical training depending on the policies of the Commanding General at Fort Leavenworth (email communication with David B. Batchelor, MS, COL-Retired, 10 July 2021). However, over half of military officers report that work commitments get in the way of exercising [21].

Although it is well-established that both exercise and dietary habits directly influence body composition, body composition alone does not directly predict health. Another useful metric is self-rated health, which is the cognitive interpretation of an individual’s physiological, psychological, and social status [22]. Among healthy US adults, self-rated health has a significant negative linear relationship with measured obesity based on BMI [23]. In previous research with the general US Army population, higher self-rated health was significantly related to lower BMI and higher APFT scores [22].

As part of a cluster-randomized clinical trial (CRCT) that primarily enrolled military officers from CGSC, we directly estimated obesity prevalence with common measurements (i.e., BMI, WC, WHtR, and BF%), and examined APFT scores by obesity status. The aim of this paper was to examine agreement in obesity prevalence for each measure and compare them with both APFT performance and self-rated health to provide a novel picture of the selected population. We hypothesized that anthropometric measures such as BMI, WC, and WHtR would underestimate obesity in comparison to BF%. We also hypothesized that obese participants would have significantly higher fail rates on the APFT and would report lower self-rated health compared to non-obese participants.

## 2. Materials and Methods

### 2.1. Study Design and Participants

This study used baseline data from a large five-year CRCT, the Army Training at High Intensity Study (ATHIS), one of the largest ever conducted to assess outcomes of the Army’s training program compared to a high-intensity functional training intervention among Army officers, an often-understudied population [24]. Participants were primarily recruited from the CGSC and Combined Arms Center at Fort Leavenworth, KS (*n* = 115) due to the installation’s willingness to collaborate with the research team and the fact that the population would remain at the same post over the study period. To meet additional recruitment needs, a smaller number of National Guard and Reserves personnel (*n* = 21, 23.8% Officers) were also recruited in Manhattan, KS. Inclusion criteria included being active duty US military personnel with no contraindications via the physical activity readiness questionnaire (PAR-Q) [25]; willingness to adhere to study protocols and assessments; and high likelihood of assignment to the same post over the 8 ± 1 month study period. Participants were excluded if they were on temporary or permanent medical profile, had any medical condition or injury preventing exercise participation, had a pacemaker or other implanted/internal electrical medical device, were on administrative leave or assigned to administrative duties, or were pregnant or lactating (women only). Participants were recruited and randomized in clusters due to the class groupings inherent at CGSC (i.e., Staff Groups who spent considerable time together in class) and the study was powered to detect a 1% change in BF%. A total of 30 clusters (or 15/condition), with an average of 5 units (soldiers) per cluster were recruited for a total of 136 participants (103 men, 33 women) enrolled over four years. Given our study sample size, we had greater than 0.8 power to detect a relationship between body composition measures of *R*^2^ = 0.063. The study received ethics approval (#7162) from Kansas State University and administrative approval from the US Army CGSC Quality Assurance Office. All study participants provided written informed consent prior to their involvement in the study.

### 2.2. Measures and Procedures

All body composition assessments (Testing Session 1) were conducted between 0500–0800 and a private assessment area or privacy screen was available, as needed. Eligible participants were emailed a copy of the consent form and pre-assessment instructions that included to drink ample fluids 24 h prior to testing; refrain from alcohol and strenuous exercise 24 h prior to testing; and refrain from consuming food, beverages, caffeine, and tobacco 3 h prior to testing. Immediately before the body composition assessment, they were asked to empty their bladder.

Participants first completed written informed consent and stood for about 5 min for proper body water distribution. After removing their shoes and socks, participant height was measured with a stadiometer to the nearest 0.1 cm. This was followed by weight to the nearest 0.1 kg, fat mass (kg), BF%, and fat free mass (kg) via the Tanita TBF-310 (Tanita, Corp., Tokyo, Japan). This single frequency foot-to-foot BIA sends a 50 Hz current through the legs and estimates the fat-free mass based on the resistance posed along the way. The body fat mass presents the difference between the body mass and fat-free mass [26]. The Tanita 310 has demonstrated strong evidence of concurrent validity for estimated BF% (r = 0.94; *p* < 0.001) when compared to dual-energy X-ray absorptiometry (DEXA) [27]. Moreover, it was reported to be a valid tool for the assessment of BF% in clinical trials [28,29]. All surfaces were sanitized before and after each participant using disinfectant wipes.

BMI (kg/m^2^) was calculated from measured weight and height. WC was taken three times to the nearest 0.1 cm using standardized landmarks with a spring-loaded tape measure [30] and was used to calculate WHtR (WC in cm/height in cm) for each participant.

Participants were emailed a link to an online Qualtrics survey (Provo, UT, USA) after completing the session. They were asked to complete the survey within the next two weeks and up to three email reminders were sent to encourage survey completion. The survey was used to collect demographic information (i.e., age, sex, race, ethnicity, and rank) and self-rated health (i.e., Would you say that in general your health is: Excellent, Very good, Good, Fair, or Poor?) for each participant. The self-rated health question was from the Behavioral Risk Factor Surveillance System and has shown high levels of reliability and validity across multiple studies [31].

Fitness assessments (Testing Session 2) were conducted between 0500–0900 at an outdoor rubberized 400 m track. Participants received the same pre-assessment instructions as for Testing Session 1. Participants completed the APFT to the Army standards at the time of the study [32]. Research assistants trained by a retired Army drill sergeant scored each participant on each event. Participants were instructed to warm-up on their own as they would for a typical APFT. They were then read the standardized instructions and given a visual demonstration for the 2 min push-up test, assigned a scorer and then timed for repetitions. After at least 10 min rest, participants were read the instructions and given a visual demonstration for the 2 min sit-up test. They were assigned a scorer and paired where the second participant secured their feet during the 2 min test for repetitions. After at least 10 min rest, participants were read the standardized instructions for the 2-mile run. They lined up on a rubberized running track and the time it took to complete 2 miles was recorded. Alternate aerobic events were not allowed.

### 2.3. Statistical Analysis

All analyses were conducted with the R 4.0.2 programming language [33]. Visualizations were created with the GGPLOT2 package [34]. For initial data visualization, scatterplots with LOESS curves (an algorithm which uses local weighted regression to fit a smooth curve through points in a scatter plot) were used to show the relationship between the body composition measures stratified by gender. Next, rates of obesity and discordance among measures were created based on recommended cut-off scores. Discordance was defined by the off diagonal of a 2x2 table comparing classification rates (i.e., percent of participants not categorized the same). The strength of the relationship was defined as no relationship (r < 0.25), weak (0.25 < r < 0.5), moderate (0.5 < r < 0.75), and strong (r > 0.75). AFPT total scores were determined with a calculator provided by https://apftscore.com/ (accessed on 8 September 2020); where fail rates (i.e., scoring less than 60 points per event or 180 points total) [32] were calculated. Finally, the relationship between obesity classification (i.e., obese versus not) based on the four body composition measures and measures of fitness and self-rated health were determined using Welch t-tests or chi-square tests, depending on the outcome.

## 3. Results

### 3.1. Demographic Characteristics

Complete characteristics of the 136 participants are presented in Table 1. Participant ages ranged from 19 to 47 years, with 87.5% 30 years and older, reflecting the demographics of the CGSC at Fort Leavenworth. Most were White, Non-Hispanic and had a military rank/paygrade of O1–O3 (junior officers; 46.3%) or O4–O10 (senior officers; 33.3%). Overall, 18.5% of the participants were classified as obese (according to BMI; 21.4% male, 9.4% female).

### 3.2. Obesity Classification Comparisons

#### 3.2.1. Relationships among Body Composition Measures

Figure 1 presents relationships among body composition measures stratified by gender. BMI, WC and WHtR demonstrated strong positive linear relationships. Moderate to strong relationships were found for all pair-wise comparisons, particularly for males. For women, the weakest relationships were found for WtHR and BF% (*R*^2^ = 0.50) and WC to BMI (*R*^2^ = 0.59) and their Loess curves suggested that the relationships were not linear throughout the distribution of these measures.

#### 3.2.2. Rates of Obesity by Body Composition Measures

Obesity percentages by body composition measure are presented in Table 2. In general, the lowest percentage of participants were classified as obese when they were assessed by BMI. This was followed by WC and WHtR, respectively, while the BF% obtained by BIA revealed the largest percentage of obese participants. These differences were particularly pronounced for women, where obesity rates were the lowest for BMI, followed by WHtR, WC, and BF%. Obesity percentages increased with age for all measures. For instance, based on categorization by WHtR, percent of obesity for those in the 40–49 age category was 4.8 times that of those in the 19–29 age category.

We examined pair-wise rates of discordance (i.e., the total percentage not categorized identically on two measures) among the different body composition measures. Discordance was lowest for WHtR vs. WC (12.6) and highest for BMI and BF% (25.2). Considering obesity as defined by BF%, WHtR showed the lowest discordance (21.0), followed by WC (23.0) and BMI (25.2), respectively.

#### 3.2.3. Obesity Classification and APFT Scores

Mean scores on the APFT by obesity classification method for non-obese and obese participants are presented in Figure 2. Those who were categorized as obese by all four body composition measures had significantly higher APFT fail rates. However, differences by classification were particularly large for BMI (mean difference = 11.1%) and WHtR (mean difference = 10.5%) defined obesity.

#### 3.2.4. Obesity Classification and Self-Rated Health

Figure 3 contains self-rated health by obesity classification. For each body composition measure, participants who were categorized as obese had significantly lower health ratings than those who were not obese. However, the relationship was particularly strong for both WHtR (*χ*^2^ = 7.34, *p* < 0.01) and BF% (*χ*^2^ = 15.0, *p* < 0.01). For BF%, participants categorized as not obese were more likely to rate their health as “Excellent” or “Very Good” (66.6%) compared to those categorized as obese (29.1%).

## 4. Discussion

This study is, to our knowledge, the first to examine obesity primarily among US military officers. This study also investigated the utilization of four measures (BMI, WC, WHtR, and BF%) to evaluate obesity prevalence in the field setting and evaluated the discordance rates between the measures. In addition, we investigated the relationships between pass/fail rates on the APFT by obesity classification and differences in self-rated health for obese and non-obese participants to examine relationships with human performance and health. As hypothesized, common measures used by the military such as BMI and WC, as well as the newer measure of WHtR, underestimated obesity when compared to BF% at least 21% of the time. Additionally, as hypothesized, participants who were classified as obese were significantly more likely to fail the APFT than those who were not obese across all obesity measurements, and obese participants had significantly lower self-rated health than those who were not obese.

Over 80% of our sample were mid-career officers. In theory, active-duty military personnel should possess good health and physical performance, and previous research has found obesity rates are significantly lower among officers than enlisted soldiers [21]. However, our results suggest that between 18.5–31.7% of all participants were obese depending on the measure used, with rates as high as 69.7% among subgroups. Except for BMI, obesity rates were higher in female compared to male participants and obesity rates were higher in each consecutively older age category. Our sample’s BMI-derived obesity rate was 5.8% higher than found by Reyes-Guzman et al. [35]. Additionally in our sample, men had 5.5% higher and women 1.4% higher obesity rates than found by Meadows et al. [21]. However, in comparison to the general adult population of the United States [36], our sample had about a 23.9% lower obesity rate overall.

Although the perception might be that BMI would overestimate obesity due to greater muscle mass among military personnel [37], in our sample BMI underestimated obesity similar to previous military research [15]. For WHtR, we found a 1.5% lower obesity rate for male compared to female participants, while the obesity prevalence was 23.1 and 39.6% lower in males than in females when they were classified using WC or BF%, respectively. However, our sample’s estimated obesity rates from BF% were 2.1, 1.6, and 1.7 times higher than those estimated from BMI, WHtR, and WC, respectively. This might be explained by the fact that BMI, WHtR, and WC are measures of anthropometric dimensions (i.e., body volume and body size), while BF% represents the relative quantity of fat tissue (i.e., defining the structure of body volume) [38]. Further, our discordance analysis showed WHtR had the lowest and BMI the highest discordance rate with BF%. Therefore, WHtR may be the anthropometric measure that most accurately estimates obesity in military officers, which is not surprising because it reflects both the size of the body frame and changes in abdominal body fatness.

Given the importance of human performance optimization to the Army, providing data about how the choice of body composition measure impacts decisions about adiposity and military readiness is a crucially important area of research. Decisions about body composition impact military personnel’s’ deployment status, opportunities for promotion, and even retention in the service. In addition, it will be important for researchers to justify their choice of body composition assessment method(s) in studies where it is an outcome—thus, our research has important implications for military health researchers. In a practical sense, anthropometric measurements are more accessible than BIA, as BIA requires power outlets, and multiple machines (that must be calibrated by a trained technician) to ensure all personnel are tested within the designated testing time frame, as well as a setting to protect the BIA devices from extreme temperatures and inclement weather. Quality BIA machines also often have high initial costs as well as incurring maintenance costs. However, if technical and financial issues are managed, BIA provides detailed information on body composition with very high precision compared to “gold standard” methods, even when core assumptions (i.e., dehydration, exercise, water and food consumption, non-voided bladder) of performing BIA measurements are violated [39].

Body size and composition are of importance for military personnel’s physical performance because they must perform tasks as efficiently and as safely as possible. Although our findings supported our hypothesis that obese participants would be more likely to fail the APFT, regardless of which obesity measure was used, previous research using BMI—which the authors regarded as a valid estimate of BF%—only found significant detriments in 2-mile run times for those who did not meet the Army’s body composition program screening standards [40]. Yet, higher BMI was positively associated with additional fitness tests including greater muscular strength, endurance, and power [40]. Other recent research found no relationship between BMI or BF% and APFT performance among Reserve Officers’ Training Corps (ROTC) cadets [41]. This echoed earlier research with ROTC cadets that found total APFT performance was not significantly associated with BMI or BF% [42]. Thus, our findings present novel fitness performance results for our study sample of mostly Army officers and add to our previous findings that greater physical activity levels were also predictive of APFT performance [24].

We also found significant differences in self-rated health for each obesity measure, with the greatest differences occurring when participants were classified by BF%. Participants whose BF% was ≤25% were nearly 2.3 times more likely to self-rate their health as excellent or very good, similar to previous research [22]. This provides evidence of a relationship between having lower adiposity and cognitively perceiving better health.

Study strengths included that our measurements to estimate obesity were measured rather than self-reported. We also completed all study assessments in the morning, matching the same time of day for all participants. However, it is of note that the calculation used by Tanita 310 is not validated in a military population nor is it publicly available for validation, which may pose certain constraints on the obtained estimation of BF%. BIA calculations should be validated in the future for this specific population. While the study sample included a higher percentage of women than the US Army as a whole (24.3% and compared to 14.4%, respectively) [21], additional women participants were needed for statistical comparisons between sexes. We were also unable to compare results by rank due to our sampling strategy. The Army has now instituted a new Army Combat Fitness Test (ACFT) replacing the APFT [41]. Health outcomes were self-rated which provides an important and much needed dimension of military personnel’s health but adding measured indicators of health would also be beneficial. This is of importance for large organizations such as the US Army and warrants further investigation.

## 5. Conclusions

This study provided novel baseline data from a CRCT among active-duty US military personnel, the majority being Army officers. Obesity rates were estimated and compared using common measures and results showed that BMI, WC, and WHtR underestimated obesity in comparison to BF% estimated via BIA. This suggests that obesity rates among military personnel may be higher than those shown by previous DoD research using self-reported height and weight, particularly among officers [2,21]. As a less-used anthropometric measure in military studies, WHtR had the lowest discordance with BF%. Our results further showed that obese military personnel were more likely to fail the APFT, particularly when classified by BMI and WHtR, reiterating the importance of body composition for human performance optimization. Future research should examine these relationships using the new ACFT. Moreover, obese participants rated their health as fair or good while non-obese participants were more likely to rate their health as very good or excellent, reflecting cognitive awareness of body composition status. Interventions to address obesity are needed, including among officers attending graduate school at CGSC to ensure they remain mission ready.

## Figures and Tables

**Figure 1 biology-11-01789-f001:**
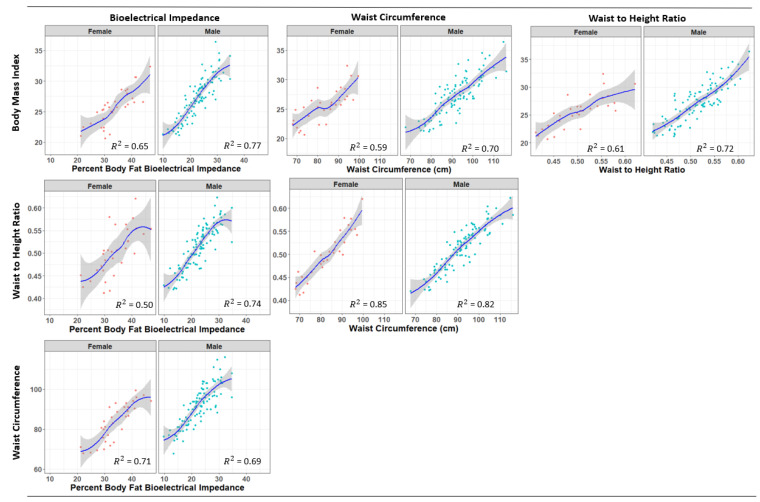
Loess curves for the relationships among measures of body composition by sex.

**Figure 2 biology-11-01789-f002:**
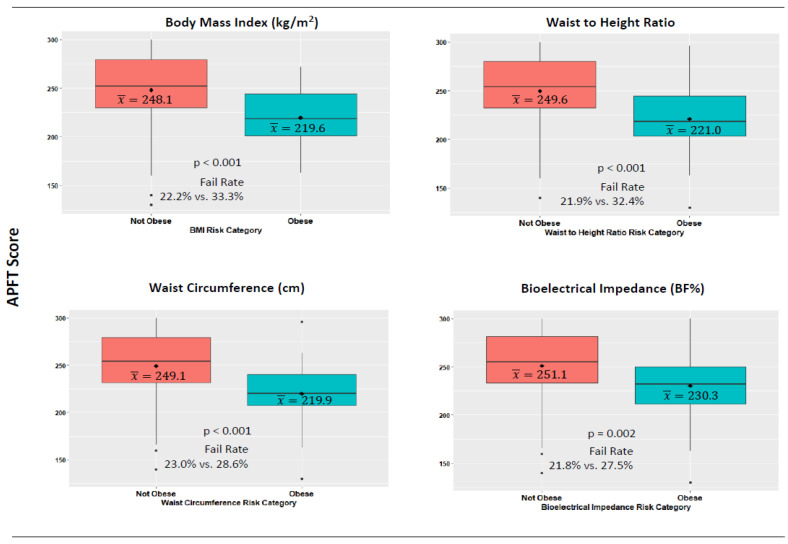
Body composition risk categories and performance on the army physical fitness test. Note: The diamond within each boxplot denotes the mean value.

**Figure 3 biology-11-01789-f003:**
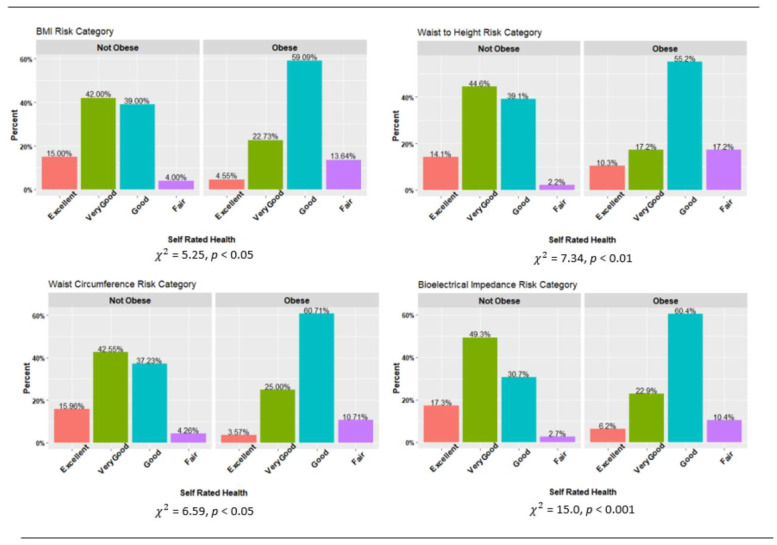
Body composition risk categories and self-rated health. Note: Chi square analyses based on dichotomizing Self Rated Health as Excellent/Very Good vs. Good/Fair.

**Table 1 biology-11-01789-t001:** Sample characteristics by sex.

Characteristics	All (*n* = 136)	Male (*n* = 103)	Female (*n* = 33)
Mean Age ± SD	34.0 ± 5.8	33.9 ± 5.8	34.2 ± 6.1
Hispanic Ethnicity (%)	4.8	5.3	3.4
Race (%) White, Non-Hispanic Black Asian Amer Indian/Alaska Native Multi Race Other	74.8 10.9 8.4 <1 1.7 3.4	78.9 8.9 6.7 1.1 1.1 3.3	62.1 17.2 13.8 0 3.4 3.4
Rank (%) E1–E4 E5–E6 Officer Trainee O1–O3 O3E O4–O10	8.1 10.6 <1 46.3 <1 33.3	7.5 10.6 1.1 50.0 1.1 29.8	10.4 10.4 0 34.5 0 44.8

Notes: E: Enlisted; O: Officer.

**Table 2 biology-11-01789-t002:** Obesity prevalence evaluated using different measures relative to sex, age, and military rank.

Sex and Age Characteristics	BMI ≥ 30	WHtR > 0.55	WC > 102 cm Men WC > 88 cm Women	BF% > 25 Men BF% > 30 Women
Sex				
All	18.5	24.6	23.0	39.7
Men	21.4	24.3	17.5	30.1
Women	9.4	25.8	40.6	69.7
Age				
19–29	11.8	5.9	11.8	17.6
30–39	19.6	27.1	23.7	40.8
40–49	19.1	28.6	28.6	52.3
Paygrade				
E1–E4	10.0	10.0	10.0	20.0
E5–E6	15.4	15.4	7.7	38.5
Officer Trainee	0.0	0.0	0.0	0.0
O1–O3/O3E	19.6	26.8	22.8	36.8
O4–O10	19.5	27.5	32.5	48.8

BMI: body mass index; WHtR: waist to height ratio; WC: waist circumference; BF%: percent of body fat; E: Enlisted; O: Officer. Note there was one Officer Trainee and one O3E in the sample.

## Data Availability

Data are available upon request from the corresponding authors.

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
