# Peer review of "Comparisons of Baseline Obesity Prevalence and Its Association with Perceived Health and Physical Performance in Military Officers"

_biology, 2022, doi:10.3390/biology11121789_

Round 1
Author Response
Thank you for your thorough review. Please see the attachment with point-by-point responses.

Reviewer 2 Report
The research article ‘Comparisons of baseline obesity prevalence and it’s association with perceived health and physical performance in military officers’ is a well written article that attempted to compare four well known body composition related methodologies as well as comparing obesity status to physical performance measures and health status.
The introduction was clear, detailed and provided a substantive level of background to readers that may not be familiar with military performance testing. The hypotheses were fair and testable relating to the scope of the data.
The methods were suitable for this type of pragmatic study using a convenience sample of military personnel. Statistical analysis was appropriate and not complicated and the results were reported in a logical fashion with tables and figures that strengthened the results section greatly. For consistency it is recommended that for Table 1., the variable ‘Rank’ be listed as Rank (%) to be similar to the other variable categories and reduce confusion.
The discussion is adequate with any relevant claims being supported by the data. Within the US Army if a soldier fails an initial height/weight (BMI) screen they are then ‘taped’ as described in the manuscript. For purposes of this study if a soldier screened over BMI standards were they then taped and if they passed tape were they then considered in the ‘non-obese’ category or was there no adjustment done for taping? This article is timely in that the military has been struggling with finding adequate, low cost, and efficient methods to measure BF% as the traditional BMI/Taping method is highly criticized. This article accentuates the difficulty in finding an efficient reliable method.
Author Response
Thank you for your thorough review, please see the attachment for point-by-point responses. Thank you.
